# Blockchain-Based Decentralized Identification in IoT: An Overview of Existing Frameworks and Their Limitations

**Seyed Mohammad Hosseini** [1,*] **, Joaquim Ferreira** [2] **and Paulo C. Bartolomeu** [3]

1. Instituto de Telecomunicações, Universidade de Aveiro, 3810-193 Aveiro, Portugal
2. Instituto de Telecomunicações, Escola Superior de Tecnologia e Gestão de Águeda, Universidade de Aveiro, 3810-193 Aveiro, Portugal
3. Instituto de Telecomunicações, Departamento de Eletrónica, Telecomunicações e Informática, Universidade de Aveiro, 3810-193 Aveiro, Portugal
* Correspondence: hosseini@ua.pt

**Abstract:** The popularity of the *Internet of Things* (IoT) and *Industrial IoT* (IIoT) has caused a rapid increase in connected entities and exposed its lack of adequate *Identity Management* (IdM). Traditional IdM systems are highly dependent on central authorities; any failure can potentially compromise the entire system. To eliminate the drawback of central authorities and evolve IdM systems to meet increasingly stringent security requirements, the *Decentralized Identification* approach has been proposed. This approach often relies on blockchain technology to provide a secure and tamper-proof method of managing and verifying identities. Therefore, this article investigates the capabilities of blockchain-based decentralized identification for the IoT domain, with an emphasis on the heterogeneity of online devices. It describes a number of features and specifications of decentralized identification with a specific focus on *Self-Sovereign Identity* (SSI), a model in which entities own their identities. The article concludes with a discussion of technical aspects as well as potential obstacles and constraints to the implementation of decentralized identification in the context of the Internet of Things.

**Keywords:** security and privacy; cyber-physical systems; secure communications; embedded devices; device security; constrained devices; Internet of Things; blockchain

## 1. Introduction

In the last few years, the Internet has experienced exponential growth with respect to the adoption of the *Internet of Things* (IoT) and *Industrial IoT* (IIoT) technologies. By 2023, there will be almost 3.6 billion Internet users worldwide [1] and businesses will increasingly rely on virtual and decentralized operating modes [2]. These trends create a new digital environment, where the core element is *identity*. A *Digital Identity* [3] is an alternative legal token (discrete identifiable unit) used for authentication and authorization, designed for the age of the Internet and corresponds to the real identity of an entity. An entity can be a person, organization, application, or device and must always be unique for being accessible and interacted with.

The use of a username and password pair, multi-factor authentication, biometric authentication, location-based authentication [4], or hardware authentication, such as *Physical Unclonable Functions* (PUF) [5], are examples of different digital identification approaches used in different scopes to assign unique identities. These approaches enable access control to the network, resources, and services, determining whether an entity is who or what it claims to be. Over the years, some of these approaches have been updated and improved, while others have become less utilized due to their increasing obsolescence.

As shown in Figure 1, digital identity systems can be classified into three types due to their basic structure: *Centralized*, *Federated*, and *Decentralized*. In centralized identity systems, such as domain name registrations or certificate authorities, all events occur in one

environment, and a central *Identity Provider* (IdP) is responsible for collecting, establishing, and managing identity information from entities. Although centralized identity systems provide great convenience to entities, managing multiple accounts between different service providers has become problematic. A *Federated Identity Management* (FIM) system is proposed as a means of enabling mutual trust between two or more centralized identity systems. This system, also known as a federated *Single Sign-On* (SSO), is an authentication scheme that allows entities to securely log into multiple applications using one identity.

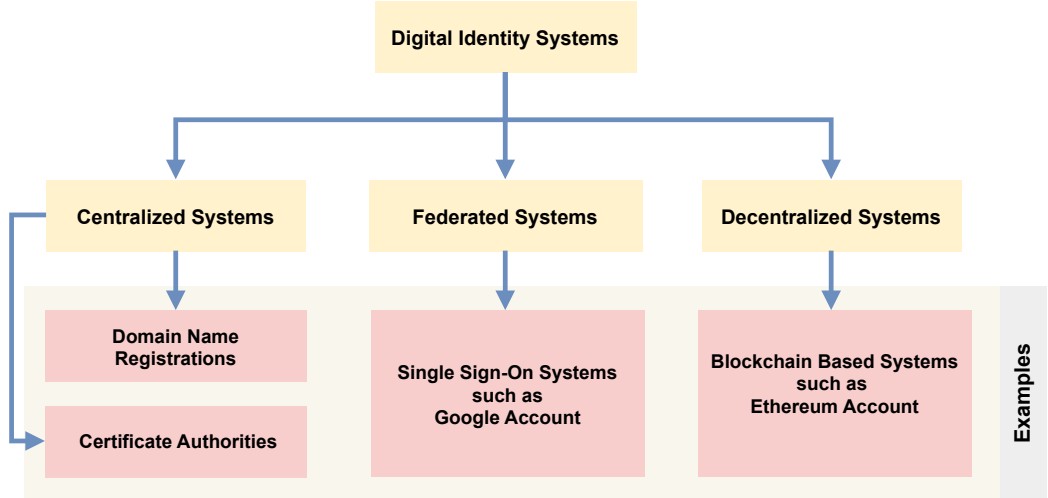

**Figure 1.** Different types of digital identity systems.

Identity management means creating, managing, and deleting the elements of the identity by the owner at any time and anywhere. The identity is existential and, therefore, its control must be with the owner. However, in centralized and federated approaches, large amounts of data are kept centrally and only a few parties or IdPs have control over them, making the entire identity system significantly vulnerable. *Decentralized Identification* based on blockchain technologies presents a new opportunity to address these concerns. It is a step forward in the realm of digital identity management, which is capable of providing all the functionality of federated identity while being independent of central authorities.

A blockchain is a distributed ledger that can be used to store and manage decentralized identity information in a secure and transparent manner. As a result of its decentralized and secure nature, blockchain is an ideal technology for the implementation of decentralized identity systems, which provide entities with control and ownership over their own identity information, while storing and verifying that information in an encrypted, transparent, and tamper-proof manner [6]. With the use of blockchain technology, identity attributes, such as name, date of birth, or government-issued identification, can be stored in a secure and decentralized manner, which allows for more efficient and secure validation of identity.

The potential for blockchain-based decentralized identification to manage identities is rapidly evolving, and a number of studies have been conducted to provide a comprehensive understanding of this emerging technology. For example, Gilani et al. [7] explores various blockchain-based identity management solutions, their limitations, and their potential applications. Another notable study by Alanzi et al. [8] presents an analysis of the strengths and weaknesses of decentralized identity management systems in terms of privacy and security. Furthermore, Alharbi et al. [9] provides an overview of the current state of the art in blockchain-based identity management for personal data, while Yang et al. [10] delves into various blockchain-based identity management approaches and their suitability for specific use cases. In general, these studies demonstrate the potential for blockchain-based decentralized identification to provide secure and efficient identity management solutions.

The purpose of our work is to provide an overview of blockchain-based decentralized identification approaches in the field of online IoT, where devices are continuously

or intermittently connected to the Internet. The main contributions of this paper are summarized below.

- A list of features and specifications for decentralized identification is provided with an emphasis on *Self-Sovereign Identity* (SSI).
- A review of advanced research in decentralized identification frameworks is presented with an emphasis on the heterogeneity of online connected devices in IoT.
- Discuss and investigate the challenges associated with the use of decentralized identification approaches in the context of digital identity.

The remainder of the paper is organized as follows. Section 2 describes the methodology of the study. Section 3 discusses the concept of decentralized identification. In addition, a review of the SSI concept is provided in this section. Section 4 analyzes the existing support structures and mechanisms, where a classification is made within the existing frameworks according to the range of target devices and environments. Section 5 examines the key challenge of adopting decentralized identification in IoT environments and provides a comparison between decentralized identification approaches and several characteristics. Finally, Section 6 concludes this article with a summary of its contributions and conclusions.

## 2. Methodology

The objective of this overview is to improve the comprehension of the reader about decentralized identification based on blockchains in the context of the Internet of Things. To achieve this goal, we outline our research questions, the data sources used to retrieve articles, the search strategy, the inclusion and exclusion criteria and the final selection as part of our overview approach.

### 2.1. Research Questions

As a first step in reviewing the literature, we identified research questions. The main question of this study is: "**Can blockchain-based decentralized identification management solutions provide a practical solution to IoT domain?**". To answer the question, three guiding questions were developed.

- What are the key technical features of a blockchain-based decentralized identification system and how do they differ from traditional IdM systems?
- What are the potential benefits of implementing blockchain-based decentralized identification in the context of IoT?
- What are the practical challenges and obstacles that must be addressed to successfully deploy a blockchain-based decentralized identification system in IoT applications?

### 2.2. Search Strategy

To conduct a comprehensive review of the literature, it is necessary to adopt an inclusive perspective. This involves selecting an adequate number of databases before starting the search, which increases the chances of identifying highly relevant articles. As part of this process, a detailed examination of the sources in *Scopus* was performed, using logic expressions for efficient search. We identified relevant articles using a set of keywords, namely: "decentralized identification", "blockchain", "Internet of Things", "self-sovereign identity", and "Identity Management". The search was refined and made more accurate using Boolean operators, including AND, OR, and NOT. We narrowed our search to article titles, keywords, abstracts, and conclusions.

### 2.3. Inclusion/Exclusion Criteria

We found that not all articles were related to the subject, so it was necessary to identify articles that matched the objectives of our study. Therefore, we define the following inclusion and exclusion criteria:

### 2.3.1. Inclusion Criteria

- Articles introducing a new blockchain-based decentralized identification method for IoT use cases.
- Since recent articles are more likely to reflect current trends in research and advances in the field, we chose to include recent articles. Therefore, research published between 2018 and 2022 was considered.
- The scope of the research was limited to conference papers and journal articles.

### 2.3.2. Exclusion Criteria

- Survey, review, and press articles that did not provide specific technological contributions were excluded from the selection.
- We excluded studies that were outdated or did not contribute significantly to the current state of the field.
- Articles that were not in English.

### 2.4. Final Selection

As shown in Figure 2, through the search, a total of 193 items were detected, of which 46 were considered appropriate for review. These 46 items comprised 22 conference papers and 24 journal articles. Despite the variety of available approaches, many of them share similarities. It was determined that five articles would be reviewed for the final review, including [11–15].

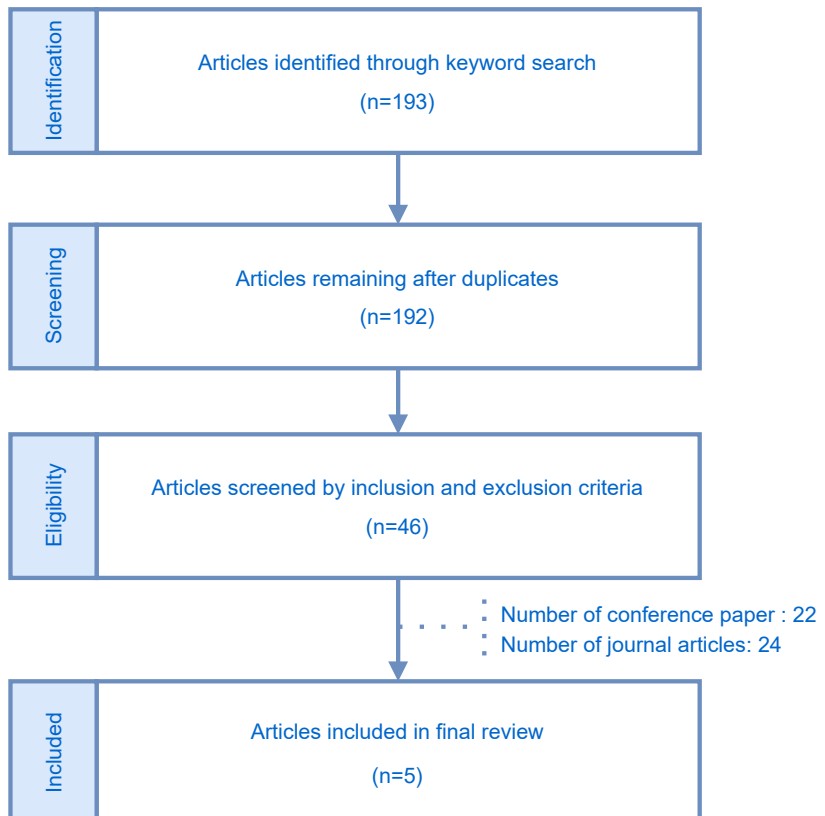

**Figure 2.** Diagram depicting the selection of studies for review.

## 3. Basic Concepts

Blockchain technologies have attracted considerable attention due to their ability to simplify trade, increase the efficiency of supply chains, improve traceability, and improve financial transactions [16]. A blockchain consists of blocks that contain details of transac-

tions that have occurred on the network. Any type of data or token exchange can be treated as transaction information. In general, there are four types of blockchain [17,18]:

- **Public Blockchains**: A public blockchain is a permissionless blockchain in which all participants can access the content of the blockchain and publish blocks. *Bitcoin* (https://bitcoin.org/en/ (accessed 13 February 2023)) and *Ethereum* (https://ethereum.org/en/ (accessed 13 February 2023)) are examples of public blockchains. To mitigate the risks associated with the use of public blockchains, a large number of anonymous nodes are needed to accommodate them.
- **Private Blockchains**: A private blockchain is a permissioned network. The entities are only allowed to join the network once their identities and other information are verified. *Hyperledger Fabric* (https://www.hyperledger.org/use/fabric (accessed 13 February 2023)) and *R3 Corda* (https://corda.net/ (accessed 13 February 2023)) are popular examples of private blockchains.
- **Consortium Blockchains**: These blockchains, also known as federated blockchains, are similar to private blockchains in that they are permissioned networks. Several organizations are participating in consortium networks that help maintain transparency between them.
- **Hybrid Blockchain**: In the context of blockchain technology, the term hybrid blockchain refers to a network that is a combination of the ideal parts of private and public blockchain solutions. In a hybrid blockchain, transactions and records are made private, but can be verified when involved, for example, by enabling access through contracts.

Several blockchains are available that support *Smart Contracts* (SC). A smart contract is a programmable application stored on the blockchain that manages transactions according to certain terms and conditions [19]. Unlike traditional contracts enforced by central authorities, smart contracts on blockchain networks do not require a third party to supervise the implementation of the terms. Several platforms, such as *Ethereum* and *Hyperledger Fabric* (https://www.hyperledger.org/use/fabric (accessed 13 February 2023)), support the development of smart contracts. Any participant can invoke the functions written in a smart contract at any time once the code is deployed on the blockchain.

As a result of the advent of blockchain technologies and smart contracts, it has become possible to create new identification systems that are decentralized. The key properties and characteristics of such systems are briefly described in the following.

### 3.1. Decentralized Identifiers (DIDs)

DIDs are a new type of identifier (a scheme with several attributes) that provides unique, verifiable, and decentralized identities. These identifiers can be cryptographically secured and are independent of any certification authority. DIDs are fully under the control of the DID subject and resolve DID documents. W3C [20] defines the DID subject as "denoted with the id property" such as a person, device, organization, object, data model, or abstract entity and the DID document as "a data set describing the DID subject, including mechanisms, such as public keys and pseudonymous biometrics, that the DID subject can use to authenticate and prove their association with the DID".

There are different types of DID structures. The complete specification can be found at W3C [21]. Below is a simple example of a DID structure for addressing a W3C DID document.

$$did : example : 123456789abcdefghi$$

In the structure, **did** is a URL schema identifier to identify a source, **example** introduces an identifier for the DID method, and **123456789abcdefghi** is the DID's method-specific identifier. DIDs can be used as a new type of *Public Key Infrastructure* (PKI). Conventional PKI is based on a central trusted party to manage identifiers, where it can become a central point of failure and compromise the integrity and security of the system. A *Decentralized Public Key Infrastructure* (DPKI) is an alternative approach to design better PKI systems. DPKI is based on DIDs and allows each entity to act as its own root authority, without relying on third parties [22].

### 3.2. Decentralized Authentication

Authentication is the process of verifying credentials to confirm identity before granting entities access to resources. Conventional (centralized or federated) authentication approaches include *Single-Factor Authentication* (SFA), such as username/password, *Two-Factor Authentication* (TFA), such as Google 2-Step Verification (https://www.google.com/landing/2step/ (accessed 13 February 2023)), or *Multi-Factor Authentication* (MFA), such as Duo Security (https://duo.com/ (accessed 13 February 2023)). However, security challenges in these authentication methods have been proven to allow attackers to steal information from entities [23].

A decentralized authentication approach is administered by a blockchain. Key-pairs are generated for an entity and the public keys are registered on a blockchain, which guarantees that the stored information is available to all participating nodes on the network and that the information is protected from modifications. In these approaches, when an entity needs to access the network, it will be authenticated on the blockchain network through a smart contract.

### 3.3. Decentralized Authorization

Authorization is part of the process of verifying which entity has access to the data, whether these systems are centralized or distributed, and whether it occurs after authentication is successful. Conventional authorization approaches, such as *HTTP Basic Authentication* and *API Keys Authentication*, are implemented in centralized or federated parties and rely on a few encryption methods to delegate authorization to access a protected resource [24]. These approaches are based on one or some of the parties and make them the target of the attack. To address this issue and provide customized access control during the authorization process, decentralized authorization approaches have been proposed, allowing the resource owner to decide *who can use the resource* (access to specific resources or all), even without trusting third parties [25].

### 3.4. Self-Sovereign Identity (SSI)

SSI is a new user-centric concept of digital identity, built on *claims* and *proofs*, to place the identity owner at the center of digital identity control without the need for a third party. Figure 3 shows the relationship between the different components of an SSI architecture versus a central/federated architecture. In SSI, the blockchain acts as a replacement for the registration authority in centralized/federated systems, called *Registry*, and the claim refers to an assertion made by an entity (person or device) that something is true. The claim is based on user-specific attributes issued by *Issuer*. The proof is cryptography evidence or an argument and demonstrates that a computational fact is true. In other words, the proof provides evidence for the claims. The true identity claim is held in a user-controlled space, typically occurring in a cryptographic network, which moves assets off the blockchain (off-chain) for privacy. A *Verifier* can compare the publicly available identifier with the identifier in the claim the user has submitted [26].

From a technical point of view, SSI is based on the use of DIDs. Since DIDs are just foundation identifiers of decentralized identity that act as an initial step in describing their subjects, W3C has published multiple drafts of *Verifiable Credentials* (VCs) to develop a framework for signing and verifying credentials and integration with DIDs. Therefore, VCs are a set of claims that cryptographically prove who issued them and allow the entity to authenticate and/or authorize another entity without the need for a central authority or third parties during the authorization process [27].

Although the term SSI remains poorly defined, this term is explained in [28], where SSI is "*an identity management system which allows individuals to fully own and manage their digital identity*". SSI allows entities to store, manage, and verify their personal identity using issued certificates. In this context, digital identity can be extended with security and privacy to IoT devices and not just to people. *Hyperledger Indy* (https://www.hyperledger.org/use/hyperledger-indy (accessed 13 February 2023)), *Veramo*

(https://veramo.io/ (accessed 13 February 2023)), *Serto* (https://www.serto.id/ (accessed 13 February 2023)), *Veres One* (https://veres.one/ (accessed 13 February 2023)), and *Jolocom* (https://jolocom.io/ (accessed 13 February 2023)) are some examples of the SSI platform, where each is characterized by a different cost (free up to a few dollars) and transaction delay (a few seconds up to hours) [27].

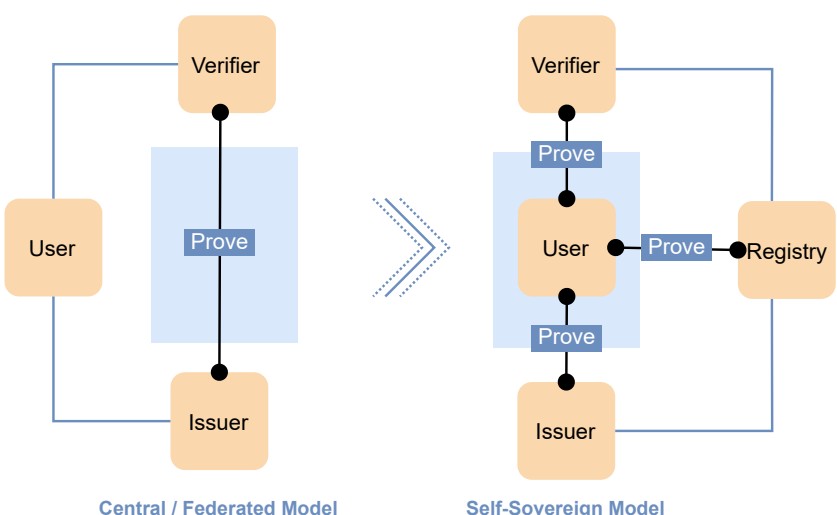

**Figure 3.** Central/Federated architecture versus Self-Sovereign architecture.

## 4. Decentralized Identity Solutions

Traditional identity solutions are challenged by the heterogeneity of IoT devices and the need for scalability of identifiers. This is especially evident when these technologies must be deployed in unreliable or constrained environments or when they need to interact with other devices in different ecosystems. IoT nodes often have limited hardware resources (memory and storage space, power, transmission capabilities, and processing power) and are not capable of processing digital identities or verifiable claims directly using common security protocols such as *Transport Layer Security* (TLS) [29]. Although efforts have been made to adapt these methods to constrained devices, they still depend on one or a few *Central Authorities* (CA), which is considered a failure point [30]. Trust in the system is undermined when CA operations are attacked by hackers or subject to government or private sector influence.

Decentralized identities have been suggested as a possible solution to address these vulnerabilities. A decentralized identity method based on DIDs and VCs provides entities with the ability to generate unique cryptographic keys that can be used to verify information and ensure the security of communications. These approaches can be classified into two categories: *relaxed* and *constrained*, reflecting the limitations of the solution's operational requirements. The first category represents the approaches and models applied in IoT scenarios where computational and connectivity resources are generally available, while the second category documents solutions targeting limited devices and environments.

### 4.1. Relaxed Operational Requirements

Devices in this category are less computationally constrained, have Internet connectivity, are not limited by battery power, and are capable of supporting stacks of communication protocols. These IoT devices are mainly controlled by traditional access control scheme models, such as *Capability-Based Access Control* (CBAC), *Attribute-Based Access Control* (ABAC), and *Role-Based Access Control* (RBAC) [31]. In these models, access rights can be granted to entities under specific conditions. However, these models are only capable of providing basic access control characteristics, and a CA is usually required to validate the access rights of entities.

For the purpose of providing a distributed and reliable access control mechanism for large-scale connected IoT devices (Microgrid), Zhang et al. [11] proposed a smart contract-based access control framework consisting of multiple contracts, named *Fast and Dynamic Access Control* (FDAC). The framework replaces the CA in traditional schema models with a private Ethereum. Within the FDAC, smart contracts provide access control methods for subject–object pairs, which can validate static access rights based on predefined access control policies, as well as dynamic access rights by assessing the behavior of the subject. Here, the subject is an entity that has access to resources, while the object is an entity that holds resources. Due to the case study, which envisages access control between multiple Ethereum nodes, the evaluation shows that the framework requires an amount of "gas" for deploying the smart contracts and storage policies between peers. This amount is equivalent to 9,500,000 in the ideal implementation of the framework, excluding the storage cost. More precisely, the concept of gas refers to the cost required by miners to execute a transaction or contract by solving complex mathematical problems, including it in a block, and adding blocks to the Ethereum. According to the amount of gas reported for FDAC, storing all policies between pairs in Ethereum and running SCs would incur significant costs. FDAC will become more expensive as the number of nodes and transactions increases.

To maximize the security of decentralized IdM and minimize data disclosure by third parties, Kassem et al. [12] introduced an Ethereum-based IdM framework similar to domain name systems, named *DNS-IdM*. Within DNS-IdM, entities can create claims and verification based on real-world identity attributes, while each attribute is validated by a specialized contract. According to the DNS-IdM architecture (Figure 4), a *Permissioned Ledger* is required to register accounts and maintain the identity information of the entities. This ledger is on a private Ethereum network and a *Permissionless Ledger* is required to store attribute-specific verification contracts. In addition, an *Inter Planetary File System* (IPFS) is connected to a permissioned ledger to store and share data in a distributed file system, and an *Identity Validation Provider* is required for validation schemes that are specifically related to a particular type of attribute. To prevent phishing, it uses *Proxy Server* to contact the identity validation provider to handle queries and set permissions to access the network. These permissions are restricted to the client's address. In the identity data ledger for third-party applications, an *Access Layer* on the service provider side provides conditional access. According to the evaluations performed on DNS-IdM, the framework requires approximately 170,000 gas only to run SCs (no information is provided regarding the amount of gas required for data storage or data recovery).

In addition to the solutions described above, other similar methods have been proposed with the aim of introducing decentralized IdM solutions [32–34]. While these solutions offer several advantages, including decentralization and data privacy, they assume that each entity has access to computational resources, is always connected to the Internet, and has direct access to distributed systems. However, some IoT devices, such as drones, robots, and low-cost computing devices, have limited processing capabilities, limited bandwidth, and limited power, and may not be connected to the Internet at all times. Consequently, these solutions are not applicable in constrained IoT environments.

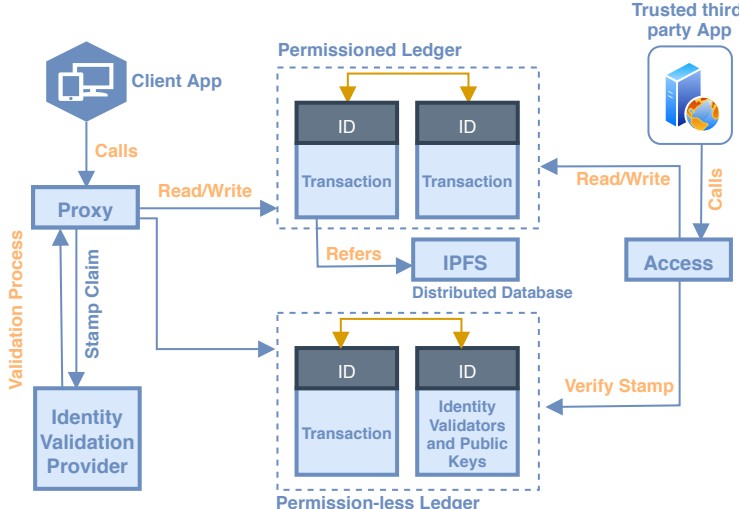

**Figure 4.** DNS-IdM architecture (adapted from [12]).

*4.2. Constrained Operational Requirements*

For the purpose of providing a method that can be used within an constrained IoT environment, *Authentication and Authorization for Constrained Environment (ACE) Working Group* (https://tools.ietf.org/wg/ace/ (accessed 13 February 2023)) developed a specific framework called **ACE-OAuth**. This framework is based on OAuth 2.0, *Constrained Application Protocol* (CoAP), *Concise Binary Object Representation* (CBOR), and *Object Signing and Encryption* (COSE) [35]. To be precise, CoAP is a lightweight web transfer protocol for constrained networks and nodes (e.g., IoT devices), which uses the *Representational State Transfer* (REST) architectural style and is typically run on top of *User Datagram Protocol* (UDP) [36]. CBOR is a binary data serialization format based on *JavaScript Object Notation* (JSON) that enables fast data transfer in a simple and concise manner [37]. COSE provides object-level layer security on the ACE-OAuth to securely transfer claims (e.g., authorization details) between entities.

ACE-OAuth introduces several flows to support the grant of approval authorization to *Clients*. A client can be any type of entity (application/device/third party) that needs access to protected resources that are hosted by *Resource Servers* (RS). ACE-OAuth flows are shown in Figure 5. In steps (1) and (2), the client interacts with an *Authorization Server* (AS) that grants an *Access Token* to access protected resources in step (3). An access token indicates the permission of a particular application to access specific parts of the user data (scope). In steps (4) and (5), which are optional and can be omitted, an RS may be configured to verify the access token by including it in a request at the AS. The authorization server's response contains some parameters, such as scope, validity, etc. If all the previous steps occur and the RS authorizes the client's request, it fulfills the request and returns an appropriate response in step (6) [35].

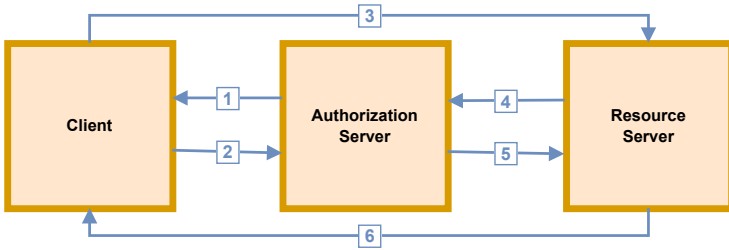

**Figure 5.** Basic protocol flow of ACE-OAuth.

Although ACE-OAuth appears to perform identification operations in restricted environments and has become widely popular, it still faces security and privacy concerns [38].

According to the implementation, ACE-OAuth is dependent on central or federal service providers for identity management, access permissions, and data storage, which means that some third parties have control over all user data that can leak or change at will, being vulnerable to hacks. To overcome these concerns, the *OAuth Working Group* has introduced a user-centered decentralized service architecture, named *Decentralized OAuth*. This service works as an alternative decentralized service architecture based on *User Managed Access* (UMA2.0). UMA 2.0 is a standard federated authorization protocol over OAuth 2.0, which allows users to control their online data from a single hub, regardless of where they are located on the Web [39]. This new service "proposes the use of a peer-to-peer (P2P) service architecture to provide service decentralization and data portability, using digital smart contracts as a legal mechanism to bind services" [40]. However, decentralized OAuth remains a proposal and no implementation or evaluation of its performance has been provided in real-world scenarios.

As discussed above, DIDs provide decentralized and verifiable digital identities. However, the results show that DID alone may not be sufficient to protect the privacy of entities in some situations [41]. Furthermore, the direct use of DIDs in constrained IoT devices has been shown to be problematic due to the overhead and resource management associated with it [42]. To ensure that DIDs and VCs are available to systems with constrained IoT devices, Lagutin et al. [13] proposed an OAuth-based user authentication method based on *Hyperledger Indy*. This method moves the processing and management of the DID and VC access policy to the authorization server. Therefore, systems with limited IoT devices can benefit from DIDs and VCs. Hyperledger Indy, which is a decentralized ledger-based identity system, has been used to create, manage, and use digital identities. According to the provided example, where an entity (lecturer) attempts to access a constrained device (printer) via a service (printing service), the entity sends an "authorization request for the constrained device (printer)" using OAuth to authorization server, including some proofs (DID is used to issue a verifiable credential). The authorization server verifies the proofs of the users and returns a *Proof-of-Possession* (PoP) access token, which allows users to communicate with the printer. This approach allows the constrained device (printer) to use DIDs and VCs even if it does not comprehend them, and the user has to disclose only the minimum amount of information (PoP access token) while being ensured that it is connected to a trusted device (based on proofs provided by the authorization server). At the time of writing this article, the cost of the proposed method has not yet been published.

A common security drawback in the ACE-OAuth framework is the inability to securely communicate and protect data over unencrypted networks and unauthenticated channels [43]. As part of the ACE framework, clients are required to establish an encrypted and authenticated channel with a trusted authorization server to securely exchange owner permissions and access tokens. Furthermore, rogue authorization servers are capable of freely issuing access tokens for any protected resource. As a solution to this problem, Alphand et al. developed a secure *End-2-End* (E2E) scheme on top of ACE-OAuth and the blockchain to provide secure authorization access to resources in restricted environments, named *IoTChain* [14]. The scheme replaces the single trusted authorization server in ACE-OAuth with a blockchain. IoTChain automatically generates access tokens for the client based on the access rights described in the SC by the resource owner if certain conditions are met. The internal storage of a SC is used to securely store the access token, rather than transmitting it to the client, as is done with ACE. It is possible for other entities to check the validity of tokens by querying the SC. Furthermore, IoTChain utilizes the *Object Security Architecture for the Internet of Things* (OSCAR) modal [44] with a self-healing group key distribution scheme to facilitate efficient multi-casting of IoT resources. The distribution of self-healing group key is a method for creating secure multi-cast communication keys for large dynamic groups of users over unreliable networks [45]. The cost of the proposed method has not yet been published at the time of writing.

Smart contracts improve blockchain technology by encapsulating policies and logic. However, the transaction cost to execute smart contracts on a public blockchain is high.

Multiple blockchains can be interconnected through interledger mechanisms, allowing smart contracts to be implemented on private or permissioned blockchains with a lower execution cost. With the use of interledger mechanisms, Siris et al. [15] proposed decentralized authorization models for constrained IoT devices based on the OAuth 2.0 framework that uses two blockchains and a multiple authorization server. According to the model, clients request access to resources through a smart contract placed on an *authorization blockchain*, while payments are made through a *payment blockchain*. Within their framework, more than one authorization server is responsible for providing authorization for the resource access request. Multi-authorization server architectures are designed to provide a higher level of resilience to node failures than a single-authorization server architecture. A key point to note is that the authorization functionality cannot be moved into the blockchain due to the fact that it involves processing secret data, such as keys used to generate token signatures and keys shared with IoT devices. The authors evaluated the proposed model using public blockchains (Rinkeby (https://www.rinkeby.io/#stats (accessed 13 February 2023)) and Ropsten (https://github.com/ethereum/ropsten) (accessed 13 February 2023)) as the payment chain and a private Ethereum network as the authorization chain to determine the cost and delay involved. Results indicate that running their framework requires 447,940 gas, while running smart contracts and mining blocks requires 44.7 s.

## 5. Discussion

Table 1 provides a comparison between several characteristics of the presented decentralized identification approaches, namely:

- **Blockchain Technology**: The use of blockchain technology has become increasingly widespread in recent years. Due to the explosive popularity of blockchain technology, various blockchain platforms have emerged, each with its own advantages and disadvantages. Since the methods presented are based on *Ethereum* or *Hyperledger Indy*, they will be discussed from a variety of perspectives in the following.

  - *Throughput*: Each blockchain has the capacity to process a number of transactions per unit of time (second), named *Throughput* or *Transactions Per Second* (TPS). This character becomes increasingly critical where the ledger is required to support an increasing number of transactions per second. Therefore, decentralized identification approaches cannot be implemented in large-scale IoT networks, such as smart cities, if transactions take a considerable amount of time. According to reports, Hyperledger's throughput is approximately 3500 TPS [46], while Ethereum is able to deliver around 14 TPS [47].

  - *Latency*: It is the time that has elapsed between the submission of a transaction and its confirmation by a blockchain network. When a device needs to transmit a large number of transactions in a short period of time, long delays can be problematic. This causes the device to run slower, affecting its performance. The latency of Ethereum has been shown to be higher than that of Hyperledger [48].

  - *Cost*: Running a smart contract on Ethereum is costly. Approaches related to this blockchain technology require a fee to execute smart contracts. A cost analysis of Ethereum is presented in [49], which concludes that "data storage in Ethereum is expensive". Regarding Hyperledger Indy, it does not have the capability to support smart contracts.

- **Scalability Evaluation**: Scalability is one of the major problems faced by blockchain technologies and refers to the size of a blockchain network and the number of entities it can support. In IoT applications, a large number of entities need to communicate through the network to identify, authorize, authenticate, share data, or possibly execute smart contracts. Evaluations have indicated that common blockchain technologies are generally not suitable solutions for managing identity and access in large-scale networks that include limited devices due to their consensus mechanisms, which need to be modified [50]. Hence, decentralized identification approaches must present an efficient solution to handle the massive amounts of data collected by a large network of

entities. A number of approaches can be employed to achieve scalability in blockchain networks, including sharding, horizontal, and vertical scaling [51]. During sharding, the data are partitioned and distributed among the nodes in order to allow parallel processing. On horizontal scaling, more nodes are added to the network in order to increase network capacity. Vertical scaling refers to adding more processing power and memory to a single node so that its performance can be enhanced. The achievement of scalability in a blockchain network requires maintaining a delicate balance between keeping sufficient computational resources available and minimizing costs associated with transaction processing. However, only Kassem et al. [12] evaluated the scalability of their solution. They examined the scalability of their system by the amount of energy consumption, and their results showed a linear relationship between cost and system operations, where the cost remains constant as the frequency of operations increases. Regarding the scalability evaluation of the presented blockchain networks, Ethereum has faced scalability issues due to its consensus algorithm (low TPS, high latency, and high cost). In contrast, Hyperledger uses a modular architecture, which permits different consensus algorithms and components to be integrated according to specific use cases. Furthermore, compared to public blockchain networks such as Ethereum, Hyperledger provides support for private and permissioned blockchain networks, which offer greater control over the network and superior performance [52].

- **Security Evaluation**: It is stated that the decentralization of the identification process can increase the level of privacy and security, allowing selective disclosure of information and protection against data manipulation [53]. Securing a network with billions of devices, where many of them have limited resources and cannot perform heavy calculations is a challenge considering that current state-of-the-art security protocols are based on a high degree of complexity [54,55]. In this regard, a practical decentralized approach needs immunity from all viable attacks and, at the same time, does not depend on heavy computing. Despite the importance of security aspects, such as privacy and anonymity, only Kassem et al. [12] and Zhang et al. [11] considered this issue in their solution and claimed to be able to avoid conventional security threats. These approaches enable authentication protection to resist different attacks and provide ownership protection by using some digital signature algorithm and keeping identities and their information in a secure and authorized ledger.

  Regarding the security evaluations of Ethereum and Hyperledger, these blockchain platforms employ different security methodologies. Overall, both platforms are robustly secured and have been thoroughly evaluated. As Hyperledger uses permissioned blockchain frameworks and focuses on enterprise security, it prioritizes enterprise security, while Ethereum provides excellent security guarantees, but it is vulnerable to attacks as it is a public blockchain platform [56].

- **Trust Evaluation**: It has always been a main topic of interest in the research community on how to promote and manage trust in distributed systems. The term trust refers to the ability of an entity to engage in an interactive relationship based on certain expectations and taking risks [57]. There is the possibility that interactions and collaborations may be unreliable due to the use of malicious behavior by some entities. Blockchain technology presents a potential alternative to transferring trust between entities to a protocol [58]. However, there are many malicious activities, such as phishing attacks, smart contract attacks, and consensus mechanism attacks, that target trust on blockchain technologies [59,60]. Furthermore, given the diversity and scale of the IoT system, trustworthiness is an important concern that must be addressed both in the design stage and during the implementation phase [61]. Therefore, to achieve a decentralized trusted identification framework, IoT must be seamlessly integrated with blockchain technology. Trust evaluation was not explicitly referenced in any of the articles reviewed.

  Regarding the trust evaluations of Ethereum and Hyperledger, although both platforms are designed to address different use cases and have different technical archi-

tectures, trust is a common element for both. The trustworthiness of Ethereum is spread among a large network of validators or miners who collaborate to maintain the credibility of the platform. However, the trustworthiness of Hyperledger is based on the credibility of its network participants who must verify their identity before being granted access to the network. Thus, the level of trust of Hyperledger is more centralized than that of Ethereum, as it relies on a smaller group of reliable participants to ensure the integrity of the blockchain [62].

- **Offline IoTs**: It should be noted that blockchain-based DID-IdM methods require online ledger operations, making them incompatible with offline IoT environments. DIDs are designed to provide proof of identity and full control over the identity in a secure and user-friendly manner between entities, regardless of whether the network is online or offline. The idea of placing SSI-based DID offline has recently been researched, but the methods are still in their infancy. For example, Alexander et al. [63] developed an offline mobile access control system based on SSI, DIDs, and VCs that uses *Bluetooth Low Energy* (BLE) to transfer data without depending on a certain transport-specific protocol. However, these offline methods raise concerns that must be addressed, including ensuring that the device is connected to the correct DID device when initializing the connection and designing a feasible data flow between entities for practical applications. Finally, none of the reviewed approaches provide a solution to the problem of serving a large number of devices in offline mode. In other words, all approaches assume that the device is connected to the Internet, even if that connection is limited.

**Table 1.** Comparison of decentralized identification frameworks.

| | Zhang et al. [11] | Kassem et al. [12] | Lagutin et al. [13] | Alphand et al. [14] | Siris et al. [15] |
|---|---|---|---|---|---|
| Blockchain Technology | Ethereum | Ethereum (permissionless and permissioned) | Hyperledger Indy | Ethereum (permissioned) | Ethereum (permissioned) |
| Scalability Evaluation | ✗ | ✓ | ✗ | ✗ | ✗ |
| Security Evaluation | Resists Against : <br> - Collusion <br> - Replay Attack <br> - Modification Attack <br> - Masquerade Attack | Protection of: <br> - Authentication <br> - Ownership | Not provided | Not provided | Not provided |
| Trust Evaluation | ✗ | ✗ | ✗ | ✗ | ✗ |
| Offline IoTs | ✗ | ✗ | ✗ | ✗ | ✗ |
| Advantage | - Dynamic Access <br> - Low time delay | - Limited Devices <br> - Privacy protection <br> - Real-world identity | - Limited Devices <br> - Dynamic Access <br> - Privacy protection | - Limited Devices <br> - Privacy protection | - Limited Devices <br> - Data Reduction |
| Disadvantage | - Costly <br> - Lack of support for limited devices | - Costly <br> - Lack of support for limited devices | - Lack of scalability <br> - No evaluation report | - Costly <br> - High delay | - Costly <br> - High delay |

The adoption of any new technology is always subject to a period of maturation and technological confirmation during which a number of requirements must be met. The adoption and use of decentralized identification in the context of IoT, in addition to its benefits, raises several concerns, which are listed below.

- **Technical**: Currently, the most significant technical challenge facing decentralized identification approaches is their infancy and inability to be widely adopted. Despite the continued efforts to add new features and resolve existing issues, none of the identified frameworks has a substantial adoption base, making it difficult to predict the impact on the scalability of billions of heterogeneous devices spread across multiple environments. In addition, there is no specific standard for writing smart contracts,

raising some concerns about data security. Smart contracts are immutable and heavily depend on developers, who can make mistakes and expose smart contracts to bugs. This makes any mistake costly for entities and allows saboteurs to steal and misuse digital assets.

- **Best Practices and Standardization**: Currently, a comprehensive and sustainable decentralized identification approach is not available for use in borderless environments for different types of entity. This implies that, without the reliance on a central authority, the entity always has access to the identification method, regardless of geography, the laws, and the type of connection. In general, efforts should be directed at consolidating the existing specifications and establishing best practices that will not only meet these specifications, but also reflect on the supporting frameworks to ensure that the GDPR requirements are met in full.

- **Central Authoritarians**: As stated above, decentralized identification eliminates the power of central authoritarians. However, governments and large corporations will be the main obstacle to these changes. Large companies and governments can try to enforce strict regulations on decentralized mechanisms, such as decentralized identification, which means that they can demand access to protected data. Furthermore, since they have financial resources, they may decide to introduce their own technology, which competes with other decentralized mechanisms that still give them an element of control. It can be concluded that decentralized identification, although well designed and implemented, will not be very effective without the support of governments and large corporations.

In general, IoT devices can be authenticated and identified securely and reliably through blockchain-based decentralized identification while preserving the privacy of entities and data.

## 6. Conclusions and Future Work

Identity is an essential aspect associated with entities such as human users, devices, and services. It should be available anytime, anywhere, and managed by the identity owner. Therefore, various identity management systems have been developed. However, these systems are highly dependent on central authorities, leading to many shortcomings, including identity theft, fraud, lack of control, and loss of data. With the intention of addressing these problems and providing persistent, immutable, verifiable, self-owned, and independent identities, the decentralized identification approach has been proposed. Within this approach, entities can create and manage their own digital identities through the use of distributed ledger technology, such as a blockchain. This approach allows entities to remain more anonymous, where they can select which information to share with others and can revoke access at any time. Thus, identity theft can be prevented and the risks associated with centralized data storage can be reduced.

Although decentralized identification offers a number of potential benefits for the Internet of Things, it still faces several significant challenges and limitations that require further work and development, including:

- The interoperability of various systems and platforms is essential. Currently, there is no interoperability between the existing solutions. Efforts should be made in the future work to achieve interoperability among various systems, so that entities can utilize their digital identities across a wide range of platforms.

- Although decentralized systems can generally be considered more secure than traditional systems, the development of new cryptographic techniques or the integration of biometric authentication techniques into decentralized systems is left to future work.

- The adoption of the new approach will be one of the biggest challenges to its success. The effectiveness of decentralized identification depends on its wide adoption and integration into a wide range of services and platforms. Efforts can be directed toward promoting the benefits of decentralized identification and making adoption easier for government, individuals, and organizations in the future.

Finally, this article examines the literature, models, and approaches and discusses the challenges faced by blockchain-based decentralized identities in the IoT domain. Based on operational requirements, the approaches presented are classified into two categories: *Relaxed* and *Constrained*. The most promising approaches in different categories are compared on the basis of several characteristics.

**Author Contributions:** All authors conceived of the presented idea. S.M.H. wrote the manuscript with support from J.F. and P.C.B. All authors discussed the results and commented on the manuscript. All authors have read and agreed to the published version of the manuscript.

**Funding:** This work is supported by the European Regional Development Fund (FEDER), through the Regional Operational Programme of Lisbon (POR LISBOA 2020) and the Regional Operational Programme of Centre (CENTRO 2020) of the Portugal 2020 framework [Project COMSOLVE with Nr. 047019 (CENTRO-01-0247-FEDER-047019)]

**Data Availability Statement:** The data used in this research article was obtained from the Scopus database. The Scopus database can be accessed through a subscription. The raw data used in this study is not publicly available due to restrictions from the Scopus database licensing agreement. However, the search query and methodology used to collect the data is available upon request from the authors.

**Conflicts of Interest:** The authors declare that the research was conducted in the absence of any commercial or financial relationships that could be construed as a potential conflict of interest.

## Abbreviations

The following abbreviations are used in this manuscript:

| | |
|---|---|
| ACE | Authentication and Authorization for Constrained |
| AS | Authorization Server |
| BLE | Bluetooth Low Energy |
| CA | Central Authorities |
| CBOR | Concise Binary Object Representation |
| CoAP | Constrained Application Protocol |
| COSE | Object Signing and Encryption |
| DID | Decentralized Identifier |
| FIM | Federated Identity Management |
| IdM | Identity Management |
| IdP | Identity Provider |
| IIoT | Industrial Internet of Things |
| IoT | Internet of Things |
| OAuth | Open Authorization |
| RS | Resource Servers |
| SC | Smart Contracts |
| SSI | Self-Sovereign Identity |
| UDP | User Datagram Protocol |
| VC | Verifiable Credential |

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
