# Peer review of "Blockchain-Based Decentralized Identification in IoT: An Overview of Existing Frameworks and Their Limitations"

_electronics, doi:10.3390/electronics12061283_

Round 1

Reviewer 1 Report

The authors compiled a review of recent decentralized identification mechanisms and their supporting frameworks in the online IoT environment. More specifically, they investigate the capabilities of Blockchain Technologies to provide robust and secure identity for the IoT, with an emphasis on the heterogeneity of online devices. The topic is of significant interest in view of the huge number of connected IoT devices.

 The authors should address the following issues in a revised version of their manuscript:

The type of paper is a Review and not a research Article. Please change.

The abstract should be more concise and avoid duplications.

Introduction

Lines 38-47 It is better to give this structural information in the form of a schematic with branches, or a Table.

Lines 69-77 Please improve the syntax of the three bullets to be consistent

Lines 231-238 More details should be given on the comparative characteristics of these SSI platforms.

Table 1: Please keep the surnames of researchers in the column tags, instead of their given names: Dimitrij should be change to Lagutin. Vasilios should be changed to Siris, Olivier should be Alphand etc.

The literature search seems not exhaustive. Also, it is necessary to cite at least a few important review papers on the general subject.

English need significant improvement. A careful reading by a native English speaker would help.

Author Response

Hi,

Thank you very much for your valuable comments. Below is the answer to the comments:

  1. Article type has been updated. Thank you for the notice.
  2. The abstract has been updated on the basis of the details provided in the work. Now, the entire article is covered in a concise manner without duplicate information.
    • The popularity of the Internet of Things (IoT) and Industrial IoT (IIoT) has caused a rapid increase in connected entities and exposed its lack of adequate Identity Management (IdM). Traditional IdM systems are highly dependent on central authorities; any failure can potentially compromise the entire system. To eliminate the drawback of central authorities and evolve IdM systems to meet increasingly stringent security requirements, Decentralized Identification approach has been proposed. This approach often relies on blockchain technology to provide a secure and tamper-proof method of managing and verifying identities. Therefore, this article investigates the capabilities of blockchain-based decentralized identification for the IoT domain, with an emphasis on the heterogeneity of online devices. It describes a number of features and specifications of decentralized identification with a specific focus on Self-Sovereign Identity (SSI), a model in which entities own their identities. The article concludes with a discussion of technical aspects as well as potential obstacles and constraints to the implementation of decentralized identification in the context of the Internet of Things.
  3. A schematic figure has been added to the article, which illustrates different types of digital identity systems.
  4. It has been improved as follow:
    • A list of features and specifications for decentralized identification is provided with an emphasis on \textit{Self-Sovereign Identity} (SSI).
    • A review of advanced research in decentralized identification frameworks is presented with an emphasis on the heterogeneity of online connected devices in IoT.
    • Discuss and investigate the challenges associated with the use of decentralized identification approaches in the context of digital identity.
  5. The discussion section has been updated. Added more details about Ethereum and Hyperledger to the article as these two platforms are mentioned in the reviewed articles.
  6. The article has been updated. The researchers' surnames are now displayed.
    1. The methodology section was rewritten.
    2. The following paragraph was added to the introduction to cover related reviews and surveys.
      1. The potential for blockchain-based decentralized identification to manage identities is rapidly evolving, and a number of studies have been conducted to provide a comprehensive understanding of this emerging technology. For example, Gilani \textit{et al.} \cite{gilani2020survey} explores various blockchain-based identity management solutions, their limitations, and their potential applications. Another notable study by Alanzi \textit{et al.} \cite{alanzi2022towards} presents an analysis of the strengths and weaknesses of decentralized identity management systems in terms of privacy and security. Furthermore, Alharbi \textit{et al.} \cite{alharbi2022blockchain} provides an overview of the current state of the art in blockchain-based identity management for personal data, while Yang \textit{et al.} \cite{liu2020blockchain} delves into various blockchain-based identity management approaches and their suitability for specific use cases. In general, these studies demonstrate the potential for blockchain-based decentralized identification to provide secure and efficient identity management solutions.
  7. The English language of the article has been revised. If the article is accepted, we will coordinate with the editor to edit the language.

Reviewer 2 Report

Dear authors,
Thank you for the opportunity to review your manuscript.

The purpose of this manuscript is to investigate the capabilities of Blockchain Technologies to provide robust and secure identity for the IoT, with an emphasis on the heterogeneity of online devices.

The references are appropriate. There are 54 references in this paper, the bibliography is recent and adequate for the research.

In my opinion, the paper addresses a very interesting and topical issue, with a very interesting analysis.

However, from my point of view, the authors must:

-explain better the final selection; it is not enough to state that „ More than 60 articles, mostly based on research databases, were collected.” They have to say how many papers were collected initially. How did they ensure they selected all the relevant articles? Did each author separately analyse all the articles and then compare the results with each other for the final election? Why did they decide to only rely on research from the last 5 years?

-develop the conclusion section.

-mention the limitations of the research, which may eventually constitute future research directions.

Author Response

Hi,

Thank you very much for your valuable comments. Below is the answer to the comments: 

  1.  Thank you for your comment regarding the selection of only five articles for the overview. We appreciate your feedback and would like to provide some context for our decision. As you are aware, the overview was intended to provide a broad survey of the literature on the topic at hand. While we agree that there are certainly more than five articles that could have been included, we felt that limiting the scope to five articles would allow us to provide a more focused and in-depth analysis of each individual article. However, we appreciate your suggestion and will take it into consideration for future wok. Thank you for your valuable feedback.

  2. The conclusion section has been updated with more data and future work as follow:

    Identity is an essential factor associated with entities such as human users, devices, and services. It should be available anytime, anywhere, and managed by the identity owner. Therefore, various identity management systems have been developed. However, these systems are highly dependent on central authorities, leading to many shortcomings, including identity theft, fraud, lack of control, and loss of data. With the intention of addressing these problems and providing persistent, immutable, verifiable, self-owned, and independent identities, the decentralized identification approach has been proposed. Within this approach, entities can create and manage their own digital identities through the use of distributed ledger technology, such as blockchain. This approach allows entities to remain more anonymous, where they can select which information to share with others and can revoke access at any time. Thus, identity theft can be prevented and the risks associated with centralized data storage can be
    reduced. Although decentralized identification offers a number of potential benefits for the Internet of Things,
    it still faces several significant challenges and limitations that require further work and development, including:
    – The interoperability of various systems and platforms is essential. Currently, there is no interoperability between the existing solutions. Efforts should be made in the future work to achieve interoperability among various systems, so that entities can utilize their digital identities across a wide range of platforms.
    – Although decentralized systems can generally be considered more secure than traditional systems, the development of new cryptographic techniques or the integration of biometric authentication techniques into decentralized systems is left to future work.
    – The adoption of the new approach will be one of the biggest challenges to its success. The effectiveness of decentralized identification depends on its wide adoption and integration into a wide range of services and platforms. Efforts can be directed toward promoting the benefits of decentralized identification and making adoption easier for government, individuals, and organizations in the future.
    Finally, this article examines the literature, models, and approaches and discusses the challenges faced by blockchain-based decentralized identities in the IoT domain. Based on operational requirements, the approaches presented are classified into two categories: Relaxed and Constrained. The most promising approaches in different categories are compared on the basis of several characteristics.

Reviewer 3 Report

The authors provide an overview of blockchain-based decentralized identification approaches in the field of online IoT, where devices are continuously or intermittently connected to the Internet. This a very nice and easy-to-read paper on a topic very suitable for this journal. My main concerts are listed below:

1- Criteria to select papers are not well defined. The papers has to include details about the searches (for instance, date).

2- I enclose a pdf with different comments to be solved.

4- More technical papers about IoT and blockchain

[1] A Decentralized Location-Based Reputation Management System in the IoT Using Blockchain. IEEE Internet of Things Journal, 9(16), 15100-15115.

[2] Internet of things. Manual of digital earth, 387-423.

Author Response

Hi,

Thank you very much for your valuable comments. Below is the answer to the comments: 

1- Regarding your concerns, the Methodology section was rewritten. 
2- After carefully reviewing your notes, here are the answers:

  • As for the typo and parentheses issues, they’ve all been fixed.
  • About the development of the Conclusion section: This section was developed after reviewing the article.

3- We appreciate your suggestion. We reviewed them. A reference to the first suggestion is made in the Introduction section "Location-Based Authentication".

Round 2

Reviewer 1 Report

can be published in the revised form

Reviewer 2 Report

The paper can be accepted in present form.

Reviewer 3 Report

The authors have solved all my main concerns, and the manuscript has been improved.